# Preparation and Properties of Graphene/Nickel Composite Coating Based on Textured Surface of Aluminum Alloy

**DOI:** 10.3390/ma12193240

**Published:** 2019-10-03

**Authors:** Linhong Xu, Ruidong Wang, Meijie Gen, Luhua Lu, Guangchao Han

**Affiliations:** 1School of Mechanical Engineering and Electronic Information, China University of Geosiences (Wuhan), Wuhan 430074, China; xulinhong@cug.edu.cn (L.X.); DivineDong@163.com (R.W.); 1201721283@cug.edu.cn (M.G.); 2Xiamen San-vision Company Limited, Xiamen 361101, China; luluhwa@163.com

**Keywords:** surface texture, composite electroplating, graphene/nickel, electrochemical corrosion, friction and wear

## Abstract

This study carried out a novel duplex surface treatment on aluminum alloy base to explore the potential improvement of wear and corrosion resistance. Regular arrayed dimple surface texture (DST) and groove surface texture (GST) were fabricated by using laser processing on 6065 aluminum alloy matrix (6065Al). Electrochemical deposition of Ni and Graphene/Ni coatings on textured surface was then performed in electrolytes with concentrations of 0, 0.5, 1 and 1.5 mg graphene. Surface morphology such as diameter of dimple and width of groove measured by C-PSCN stereo microscope presents addition of graphene helps to refine and homogenize the coating. Corrosion resistant properties of the duplex surface treatment were examined by electrochemical corrosion tests and wear resistant properties were tested by UMT-Tribo Lab friction and wear tester in a dry sliding condition at room temperature. Electrochemical corrosion tests results show that the corrosion resistance of samples is related to the specific surface texture and the dimple texture can improve the electrical corrosion parameters, such as the electrode potential, greatly. Friction and wear tests show that the textured Gr/Ni electroplating coating with the 1.5 mg graphene content has best wear properties under vertical friction and each index, such as the coefficient of friction and wear trace width, are superior to other conditions of samples.

## 1. Introduction

Aluminum alloy has been considered a good choice as a material in the aviation, aerospace, automotive, marine and chemical industries, due to its excellent specific strength and plasticity, high electrical conductivity, thermal conductivity and remarkable corrosion resistance [1]. Yet, like most of aluminum alloy, 6065 aluminum alloy still displays poor wear resistance and corrosion resistance by Cl-ions when exposed to neutral aqueous solution [2], which greatly restrained its lifetime and the application area. Recent research have found that the addition of materials such as SiC, CNT-carbon nanotubes, nano-diamond and MoS_2_-nanosheet to the composite plating solution results in a certain improvement in corrosion resistance and friction resistance. However, difficulty still arises with the electroplating process is the formation of natural oxide film on the surface of the aluminum alloy, hindering the direct reduction of metal ions on the surface of the aluminum alloy to obtain a plating layer with ideal adhesion situation [3]. Therefore, a conductive additive necessary to increase the rate of electrodeposition should be considered to achieve a more compact bonding of the composite electroplating layer on the surface of the aluminum alloy.

Graphene, with excellent electrical, thermal, chemical and mechanical properties, has gained extensive attention since its advent. Single-layer graphene is a planar thin film material which composed of carbon atoms in sp2 hybrid orbitals [4,5]. This unique structure not only has outstanding electrical conductivity, very low electrical resistivity and extremely low electron migration speed, but also has extremely high strength and very good thermal conductivity [6]. At the same time, graphene nanoplates, with a few layers exhibit similar properties as single layer graphene, are much easier to produce, store and handle than single layer graphene, which makes it as perfect nanofillers to reinforce conventional structural engineering materials and endow them with some new functions [7]. 

Previous studies have demonstrated that graphene can be used as an additive for electroplated composite. Szeptycka [8] finished composite electroplating in Watts nickel plating type to increase the effect of wear resistance for the Ni-graphene electroplating layer, the abrasion resistance relative to the composite having 3.7 to 6.9 times the pure nickel coating layer. Chen [9] performed graphene/nickel composite plating on 45 carbon structural steel, and found that the thickness and hardness of the coating increased with the addition of graphene, and the surface roughness decreased with the increase of graphene addition. Jiang [10] studied the anti-corrosion of mild steel composite electroplated with Ni/graphene; the result of the electrochemical corrosion test was found to be −234.6 mv, much higher than the original matrix’s corrosion potential of −579.4 mv and pure nickel coating’s -266.5 mv. This is the reason that graphene can prevent the occurrence and development of defect corrosion at these locations in cracks, gaps, and micropores in the Ni matrix [11]. All this research has shown graphene/nickel composite electroplating has great significance in increasing the service life of aluminum alloys due to corrosion and friction and wear.

With the development of textured surface science, people now begin to re-recognize the wide application of non-smooth surfaces in various fields. Among of them, surface texture (ST) has been confirmed as an effective and economical surface treatment technique that can be applied to a great range of materials and it has been attached greater importance in various engineering fields. Ta [12] utilized nanosecond pulsed fiber laser to texture brass plates in parallel microchannels. Such a surface may be used in a chemical sensor for accurate measurement of chemical concentrations. Zhang [13] fabricated micro-pit array texture on TC4 titanium alloy surface by YAG-T80C laser marking machine. This kind of titanium alloy with micro-pit texture can bind tightly to human body and improve the wear resistance of titanium alloy in vivo. Bonse [14] obtained periodic uniform ripple texture on Ti6Al4V titanium alloy surface by femtosecond laser pulse irradiation. Compared with untreated surfaces, the reciprocating coefficient of friction between corrugated textured surfaces and hardened steel balls under engine oil lubrication is lower and more stable. Shen [15] presented a route to fabricate a robust anti-icing superhydrophobic surface containing the hierarchical structures of microscale array patterns (constructed by micromachining) and nano-hairs (prepared via hydrothermal growth) on the Ti6Al4V substrate. Ti6Al4V surface exhibited a higher anti-icing potential. It caused a longer icing-delay time (approximately 765 s) to hinder the ice formation and growth at -10°C, and the ice adhesion strength was also only 70 kPa, which is much lower than those on the other sample surfaces (700 kPa for polished surface). 

However, neither the surface texture nor the surface coating can solve the corrosion resistance and wear resistance of aluminum alloy well [16]. A combination of surface modification and surface texturing is considered to make use of the inherent positive effects of the surface modification layer as well as that of surface texture. Xiang [17] focused on the preparation technology and characterization of micro-texture diamond-coated tools with advanced cohesive and wear-resistance parameters. Microtextures were processed on the rake face of carbide milling inserts (YG6, WC-6% Co) by laser marking machine, and diamond films were deposited on the substrates by the hot filament chemical vapor deposition. The experimental results of milling tests show that the microtextures can enhance the adhesion between the diamond films and the tool surfaces. The microtexture diamond-coated tools have a longer tool life and a better cutting performance than uncoated tools and conventionally coated tools. Anwer [18] developed the textured composite surface with protruding fibers, which exhibits an extremely high coefficient of friction on ice. This kind of novel composite material with improved wear resistibility is aimed to determine with the target to maintain its slip-resistance properties over extended use. Shajari [19] fabricated Fe–Al_2_O_3_ composite cladding on AISI 1040 substrates by the cost-effective process of gas tungsten arc welding. The microscopic images showed coaxial fine grains in the weld composite surface, there are no cracks in the surface layer created and Al_2_O_3_ particles are distributed uniformly in the surface layer. The purpose of the research could achieve optimal parameters of claddings in an inexpensive method. Liu [20] prepared hydrophobic microstructure magnesium alloy surface by laser combined with electrochemical deposition. After 3.5 wt% NaCl electrochemical test, it was found that the coating had a certain corrosion resistance, and its corrosion potential increased by more than 0.2 V.

Thus far, however, there has been little discussion about duplex treatments of surface modification-surface texturing for 6065 aluminum alloy matrix (6065Al). Inspired by bionic studies and taking the advantages of surface texturing and surface modification, in the present work, the surface texturing/composite electroplating (ST/CE) duplex treatments were applied on 6065Al. Regular arrayed dimple surface texture (DST) and groove surface texture (GST) were adopted and fabricated by using laser processing on 6065Al matrix respectively and electrochemical deposition of Ni and Gr/Ni coatings on textured surface was then performed in electrolytes with concentrations of 0, 0.5, 1 and 1.5 mg graphene. The influences of different texture and content of graphene on friction behavior and corrosion resistance were investigated by micrographs, electrochemical experiments and friction and wear experiments.

## 2. Experimental Procedures

This section presented procedures of the duplex treatments of surface modification-surface texturing on 6065Al matrix. The scheme of this processing is shown in Figure 1. Here, Texture surface was fabricated by using laser processing and then the graphene/Ni composite electroplating processing was carried out on the textured surface.

### 2.1. Materials

The material selected for the present research work is Aluminum 6065 alloy, which is commercially available with dimension of 70 mm × 20 mm × 2 mm under cold-rolled condition and its composition is given in Table 1. Surface of 6065Al plate were ground by grit papers for 10 min in order to remove oxide and obtain a smooth surface.

### 2.2. Surface Texture Construction

A P50Q pulsed fiber laser (Wuhan Raycus Fiber Laser Technologies CO, LTD, Wuhan, China) with 1060 nm wavelength was employed here to fabricate the needed surface texture. Referring to other researchers, dimple and groove surface texture were adopted, which are based on the two typical wear resistant and corrosion resistant surfaces of lotus leaf and shark skin [21,22]. Detailed parameters were listed in Table 2. Laser processing parameters used were as follows: laser power of 45 W, scanning speed of 200mm/s, scanning time 10 min. After laser treatment, the hardness of substrate (41.4 HV) can be improved to HV 102.2 HV after laser processing, which benefits the enhancement of the properties of the textured substrate (as shown in Figure 2).

### 2.3. Electrodeposition Process

In this experiment, an Ni plate (purity 99.9%, Yingxin Metal Material Co, Ltd., Wuhan, China) with a surface of 20 mm × 20 mm was used as the anode for electrodeposition. The cathode, with distance of 60 mm towards the anode, was a textured 6065Al plate with size of 20 mm × 20 mm. To improve the adhesion of coating, both electrodes were firstly immersed in 75% anhydrous ethanol and deoiled by ultrasonic heating with a PS-20A Digital Ultrasonic Cleaner (Jeken Ultrasonic Cleaner Ltd., Wuhan, China) for 10 min. Then, the nickel plate was pickled in 3% hydrochloric acid for 15 min, and the textured sample plate was soaked in NiCl_2_ saturated solution and 37% HCl mixed solution for 50–60 min at room temperature to remove surface rust. Electrolyte [23], as can be seen in Table 3, was prepared by adding nickel ammonium sulfamate tetrahydrate (350 g/L), nickel chloride (10 g/L), boric acid (20 g/L), citric acid (5 g/L) and sodium dodecyl benzene sulfonate (0.015 g/L) into 450 ml deionized water and then stirred by JJ-1 stirrer (XinRui Co, Ltd. China) for 10 min in a water bath at 40 ℃. The pH value of the solution obtained was 3-4.

Here, multi-layer graphene aqueous solution was used supplied by Nanjing Ji nano company produced by using physical method (Purity of the graphene higher than 99%, thickness < 0.6nm, electrical conductivity > 1000 s/cm). The composite plating solution was obtained through adding 1 ml with different concentration graphene aqueous solution (0.5, 1 and 1.5 mg/ml) for comparing into nickel electrolyte and stirred sufficiently by ultrasonic vibration. Composite graphene electroplating were then started. During electro-deposition process, both the anode and the cathode were connected to the MS-155D electroplating power supply (MAISHENG Co, Ltd., Wuhan, China) and a constant current power with 13.75 A/dm^2^ was applied. The electroplating time is 5 min at 40 °C and the stirring speed is 300 rpm. The samples were then washed with deionized water after electroplating process and finally put into the SX2-4-10 electric furnace (Yahua Co. Ltd., Wuhan, China) at 200 °C for 0.5 min for drying. Detailed electroplating parameters are shown in Table 3.

### 2.4. Characterization

#### 2.4.1. Surface Morphology

Surface morphology such as diameter of dimple and width of groove were measured by C-PSCN stereo microscope (Nikon Corporation. Japan). Scanning electron microscopy (SEM) images of the samples were analyzed using an Su 8000 FE-SEM (Hitachi, Ltd, Tokyo, Japan) and the amount of electrodeposited graphene was estimated by using an energy dispersive spectroscopy (EDS-550I) which was attached to the FE-SEM. The sectional images of the textured electroplating coatings were captured by MA100N-optical microscope (Nikon Corporation, Tokyo, Japan).

#### 2.4.2. Corrosion Resistance

Electrochemical corrosion tests were carried out on CS2350 double channel electrochemical workstation (Wuhan CorrTest Instruments Corp. Ltd., Wuhan, China) equipped with three-electrode system. Reference electrode and auxiliary electrode are saturated calomel electrode and platinum electrode, respectively. Sample obtained above was used as the working electrode. Results of the electrochemical corrosion test were obtained through the corresponding software CS Studio 5. Tafel lines were recorded at a scanning rate of 1 mV/s, which is low enough to have a good response for the system, from 0.8–0.2 V versus open circuit potential. Electrochemical impedances were obtained in the frequency range of 0.01 Hz–10 kHz and the amplitude of sinusoidal signal was 10 mV [24]. These tests were carried out in neutral 3.5% NaCl solution at room temperature [25].

#### 2.4.3. Friction and Wear Resistance

UMT-TriboLab friction and wear tester (Bruker Nano, Inc., Campbell CA 95008, USA) was employed to investigate the tribology behavior of coatings in a dry sliding condition at room temperature. A Si_3_N_4_ ball with the diameter of 6 mm was selected as a counterpart. Test parameters are as follows: a normal load of 10N, sliding velocity of 5 mm/min, reciprocatory displacement of 10 mm and test time of 10 min. The worn surface was observed and examined for analyzing.

Here, to explore different texture surface influence on the friction result, two friction directions (parallel and 45° to dimple arrangement) for dimple texture surface were tested, referred to as D0 and D45 in the following Figure 3, table and chapter. Three friction directions (parallel, 45° and vertical to the groove direction) for groove texture surface were tested, referred to as G0, G45 and G90, respectively, in the following Figure 3, table and chapter.

## 3. Experimental Results and Discussion

### 3.1. Morphologies of the Coatings and Textured Surface

Figure 4 and Figure 5 are surface images observed, respectively, for DST or GST of 6065Al plate obtained under different conditions in order to have a better comparison and investigation. 

The average diameter of the original dimple shown in Figure 4a is around 200 μm and the diameter in both in Figure 4b,c was reduced to be around 160 μm. Similarly, the width of the groove surface texture in Figure 3a is about 200 μm, and the width in both in Figure 5b,c increases to about 260 μm. It was found that the pit diameter and groove width of the samples are in good agreement by measuring several different positions on the surface of the sample, which indicates that the coating is continuous and uniform. As can be seen from Figure 6, sectional images of the GST electroplating coatings show the electroplating coatings and the alloy matrix are uniformly and tightly connected. Compared with the textured nickel coating, the GST graphene/nickel composite coating has better bonding with the Al alloy substrate, and there are no obvious impurities and defects at the interface. The uniformity of the pit diameter and groove width on the surface of the sample is very good by measuring multiple positions of the sample, which shows that the coating is continuous and uniform. During the composite electrodeposition, Ni^2+^s would be preferably electrodeposited onto the graphene and lead to the formation of microbulges on the electroplating coatings. In the process of composite electrodeposition, Ni^2+^s would be preferentially electrodeposited onto graphene and encapsulated graphene which finally lead to form micro-convex bulges on the electroplating surface. This phenomenon can be found in Figure 6c and Figure 7f and similar observations were also reported by Jiang [10]. Thus, the surface morphology of the composite electroplating coating with the different contents of graphene is probably correlated to the electric force between the composite and the Ni matrix during deposition, which influences the absorption of composite on the surface of the electrodepositing layer [26]. The EDS result in Figure 7g gives that the content of C and Ni are estimated to be 3.496% and 64.214%, respectively, which indicates that graphene were successfully adsorbed on the graphene/Ni composite electroplating coating during electrodeposition, which could greatly increases the number of nucleation sites for Ni^2^+ to reduce, resulting in a fine grained and intact arrangement of Ni crystals [27]. The chlorine is shown in the EDS and it may be introduced by nickel chloride during electroplating.

Furthermore, by comparing Figure 4b,c or Figure 5b,c, it is obvious to be found that the surfaces of (c) are much more smooth and uniform than those of (b). The surface in both Figure 4b and Figure 5b is a little bit rough and there is an obvious bulge on the surface. A possible explanation for this might be that graphene can effectively inhibit the growth and coarsening of grains, and hence, refine the grains of coated metals. Another possible explanation might be that the strong conductivity of graphene can also promote the uniform distribution of current on the surface of the cathode, and the deposition rate of nickel ion in the bath is also more stable [28]. Both factors make graphene composite electroplating has obvious effect of densification on the nickel plating surface, thus, the coating of composite graphene/nickel electroplating presents much more smoothness and uniformity.

### 3.2. Corrosion Resistant Properties of the Electroplating Coatings

Figure 8 and Figure 9 are Tafel curves of electroplated 6065Al, respectively, for dimple and groove surface texture in 3.5% NaCl test solution. Table 4 gives the corrosion potential and current obtained by fitting Tafel curve under different conditions. 

As can be seen from Figure 8 and Table 4, the corrosion potential and corrosion current density for DST samples are −655.44 mV and 469.92 μA/cm^2^, respectively. For DST/CE(Ni) samples, the corrosion potential increases to −618.63 mV, which means the corrosion resistance of 6065Al sheet is improved to a certain extent. Furthermore, there was a significant positive correlation between the corrosion resistance and graphene content from test result. The DST/CE(Ni+1.5Gr) get the most notable improvement of corrosion resistance. The corrosion potential of the composite coating moves forward to −601.69mV, and the corrosion current also decreases by 13.90% to 404.58 μA/cm^2^. Meanwhile, the corrosion test parameters of stripe textured composite electroplating coatings also appeared a similar trend. This result is consistent with another study [10], in which the addition of graphene played a much more obvious role in improving the corrosion resistance during the electrodeposition of Ni.

Quite possibly, these results are due to the excellent electrical conductivity of graphene that make the nickel have more nucleation points on the surface of the coating. The metal particles crystallized at these nucleation points will deposit at a faster rate, thus, the nucleation rate of the composite electroplating coating is higher than that of the pure nickel electroplating coating. During the process, graphene can effectively prevent the growth of metal grains, thereby refining the grain. Therefore, the addition of graphene composite electroplating coating can obtain a much more compact coating reflecting a better anti-corrosion effect.

Further analysis with data in Table 4 showed that DST/CE (Ni+Gr) has a higher corrosion resistance than GST/CE (Ni+Gr). The minimum corrosion potential of the composite coating with DST can reach −601.69 mV, which is 4.45% higher than that of the GST. This result may be explained by the fact that for dimple surface texture, a single pit completely immersed in the corrosive solution can easily form a closed small cavity, and the air filled in the cavity can reduce the contact area between the liquid and the coating and hence reduce the possibility for harmful ions such as Cl^-^ ions to reach the matrix of dimple surface texture [29]. On the contrary, for GST, it is difficult to form a closed space for air on the groove texture surface under same corrosion condition. Hence, the corrosion potential of the dimple texture composite electroplating coating will be lower than that of the groove texture composite electroplating coating, and its corrosion resistance is also improved. The corrosion current density of GST/CE (Ni) is less than that of DST/CE (Ni+1.5Gr) because the two values are too close, so the corrosion resistance of dimple textured composite coatings is more explained by the comparison of corrosion potential.

The corrosion resistant properties of textured graphene/nickel composite electroplating coatings were evaluated by the electrochemical impedance tests. The Nyquist plots of all the surfaces with different treatments were shown by Figure 10. Usually, the larger diameter of capacitive loop in Nyquist plot implies higher corrosion resistance [30]. From Figure 10, we can find that DST/CE (Ni+Gr) coatings exhibit remarkable higher impedance compared to GST/CE (Ni+Gr) coatings and the impedance of coating increased with higher graphene content. These results are consistent with that of potentiometric polarization curve analysis. DST/CE (Ni+Gr) coating have the highest impedance and better corrosion resistance in plating bath containing the largest amount of graphene addition.

### 3.3. Friction and Wear Properties of the Electroplating Coatings

#### 3.3.1. Coefficient of Friction

Friction coefficients of textured surfaces coated with different electroplate composites under stable friction and the variation trend of COF on the surface for all the samples are shown in Figure 11 and Table 5.

From Figure 11 and Table 5, it can be seen that under the same friction conditions, the friction coefficients of ST are higher than those of ST/CE (Ni) and untreated base material (1.353), which shows that the wear resistance is obviously improved after pure nickel plating, at same time, the friction coefficient of textured graphene/nickel composite coatings decreases gradually with the increase of graphene content. The friction coefficient reaches the minimum and stabilize at 0.735 when graphene content is 1.5 mg with decrease rate 29.74% and 45.67% by comparing with ST/CE(Ni) and untreated 6065Al. The explanation for this phenomenon is due to the lubricating effect of graphene itself. During the friction and wear experiments, graphene forms a lubricating film on the surface of the coating, which reduces the friction and wear of the coating surface and minimize the wear rate of the coating.

By comparing the friction coefficients of dimple and groove surface texture coatings, it can be seen from Figure 10 that the wear resistance of groove texture coatings is better than that of dimple texture coatings, and the friction coefficients are basically lower than that of dimple texture coatings. It is highly possible that interval spacing of the groove could decrease the accumulation of deformation to reduce the stress concentration and hence more energy is needed for crack propagation in groove texture [31]. When subjected to friction, the resulting cracks are blocked by the groove spacing but can easily bypass the pit. Therefore, under the same friction condition, the friction coefficient of the dimple texture is higher than that of the groove texture coating.

Besides this, by comparing the frictional coefficients of the groove texture surfaces under different frictional directions, it can be found that the lower the coincidence degree between frictional direction and stripped direction is, the more obvious the decrease of frictional coefficients is. When the frictional direction is perpendicular to the groove direction, the frictional coefficient is the lowest, which is 0.735. This is due to the spacing of the groove isolates the propagation of cracks and limits the increase of friction coefficient on the textured surface of the fringes [32].

#### 3.3.2. Wear Trace Width

All the micrographs of those worn samples surface after the test are shown in Figure 8 and Figure 9, which gives the trend of wear trace width for all specimens under different test conditions.

Figure 12 gives 25 images for the textured surfaces obtained with different textured composite electroplating coating to compare the worn trace width of the samples. 

Comparing with the friction scratches of all the specimens in Figure 12 and Figure 13, we can see that the distribution of scratches becomes more and more sparse with the increase of graphene content, and the distribution of scratches becomes more and more average compared with the bionic surface of aluminum alloy matrix, while the length and width of the micro-scratches become smaller and smaller. This shows that the graphene layer mixed in the coating has a good anti-wear effect. Because of the strong conductivity of graphene itself, metal particles can be more easily aggregated on the surface of the coating, which plays the role of increasing nucleation points. The nickel metals gathering around graphene will gradually synthesize micro-convex hulls, which reduce the actual wear area in friction, so the friction effect is delayed and the friction loss is reduced accordingly.

It is also possible to verify that graphene reduces friction by comparing the trend of worn trace width on composite coatings with different graphene additions in Figure 12. Generally speaking, the worn trace decreases with the increase of graphene content under various friction conditions. The biggest decrease is that the groove sample with 1.5 mg graphene added in the vertical friction direction decreases from 1.925 mm to 1.303 mm, with a decrease of 32.31%. This apparent decrease in worn trace width can be explained by the micro-bulges of the composite electroplating coating with 1.5 mg addition content of graphene and the groove textured composite surface has the best anti-friction property perpendicular to the striate direction. The results of Figure 13 also show that when graphene content is 0.5 and 1.0, the width of friction marks in parallel fringes is lower than that in vertical fringes. This is because the surface will be more uneven when vertical fringes are rubbed. As can be seen from Figure 9a,c,e, the real-time friction coefficient jitter of vertical fringes is greater in the first 100 s. It will make the coating adhere to the friction material during the friction process, which will lead to the increase of the friction loss. However, in general, with the increase content of graphene, the groove textured graphene/nickel composite electroplating coating exhibits an optimum wear resistance when rubbed in the direction of vertical stripes.

## 4. Conclusions

The following conclusions can be drawn from the present study:(1)Graphene/Ni composite electroplating coating on groove and dimple textured 6065Al surface were prepared in this paper. Sectional images of the samples show the electroplating coatings and alloy matrix are connected uniformly and tightly. Electrochemical corrosion test results show that the corrosion resistance of the dimple textured graphene/Ni composite electroplating coating is better than that of the groove textured one due to the effect of those small air pockets created by dimple textured surface.(2)Friction and wear experiments revealed that the surface friction properties are related to the texture, the amount of graphene added and the direction of friction. Anti-friction performance of the groove textured graphene/Ni composite electroplating coating is higher than that of the dimple textured one.

## Figures and Tables

**Figure 1 materials-12-03240-f001:**
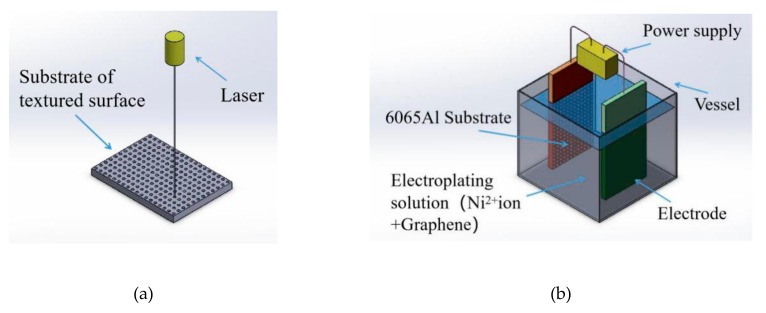
Surface texturing/composite electroplating (ST/CE) duplex treatment processing based on 6065Al. (**a**) Laser process. (**b**) Electrodeposition process. 6065Al: 6065 aluminum alloy matrix.

**Figure 2 materials-12-03240-f002:**
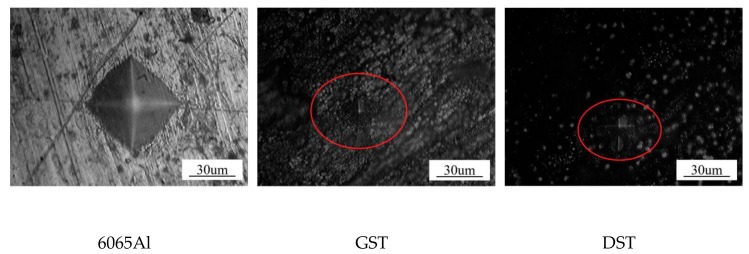
Micro-hardness Indentation on the surface of different samples.

**Figure 3 materials-12-03240-f003:**
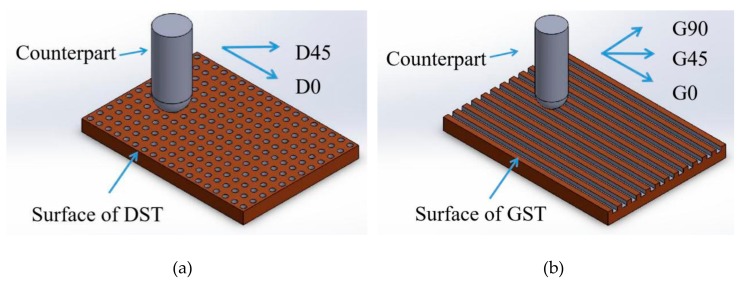
The principle diagram of the friction and wear experiments. (**a**) Friction direction on DST samples, parallel (D0)and 45° to dimple arrangement (D45), (**b**) friction direction on GST samples parallel(D0), 45° (D45) and vertical to the groove direction(D90).

**Figure 4 materials-12-03240-f004:**
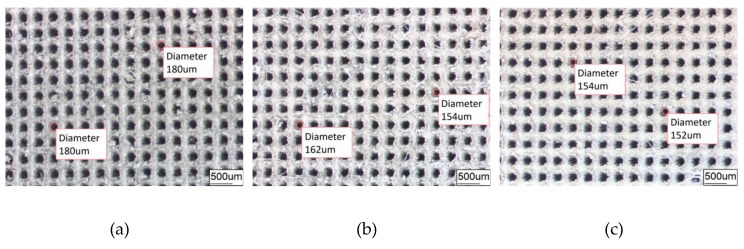
Dimple surface texture morphology obtained under different conditions. (**a**) DST samples; (**b**) DST/CE (Ni) samples; (**c**) DST/CE (Ni-1.5Gr) samples.

**Figure 5 materials-12-03240-f005:**
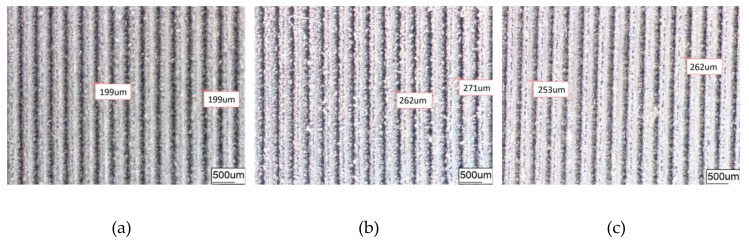
Groove surface texture morphology obtained under different conditions. (**a**) GST samples; (**b**) GST/CE (Ni) samples; (**c**) GST/CE (Ni-1.5Gr) samples.

**Figure 6 materials-12-03240-f006:**
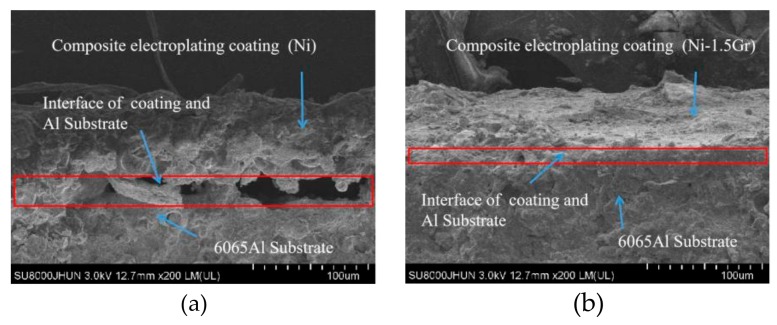
Sectional images of the textured electroplating coatings. (**a**) Composite electroplating coatings (Ni). (**b**) Composite electroplating coatings (Ni-1.5Gr).

**Figure 7 materials-12-03240-f007:**
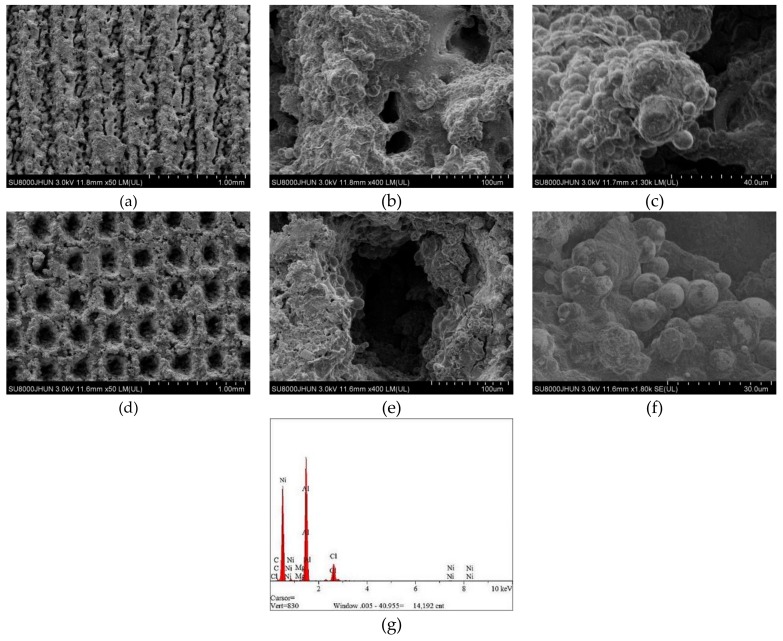
SEM images of the textured graphene/nickel composite electroplating coatings. (**a**–**c**) GST/CE (Ni-1.5Gr) samples. (**d**–**f**) DST/CE (Ni-1.5Gr) samples. (**g**) Energy dispersive spectroscopy (EDS) result of Gr/Ni coating

**Figure 8 materials-12-03240-f008:**
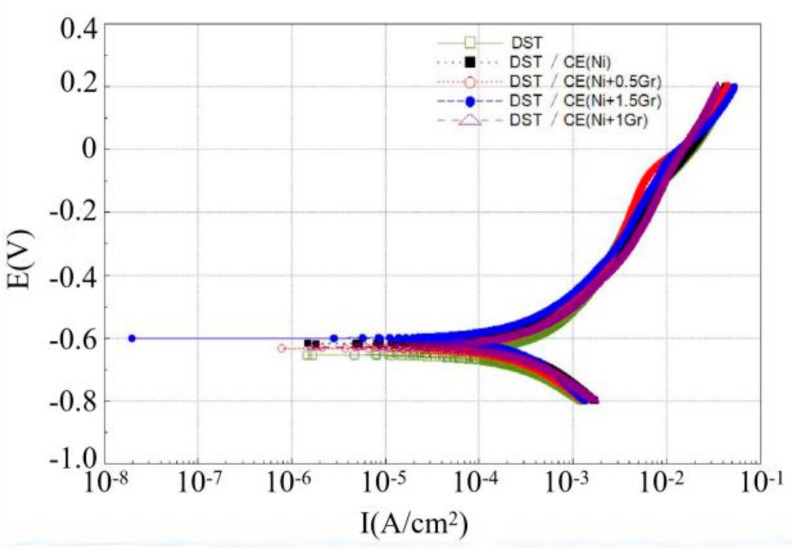
Tafel curve of dimple surface texture obtained under different conditions.

**Figure 9 materials-12-03240-f009:**
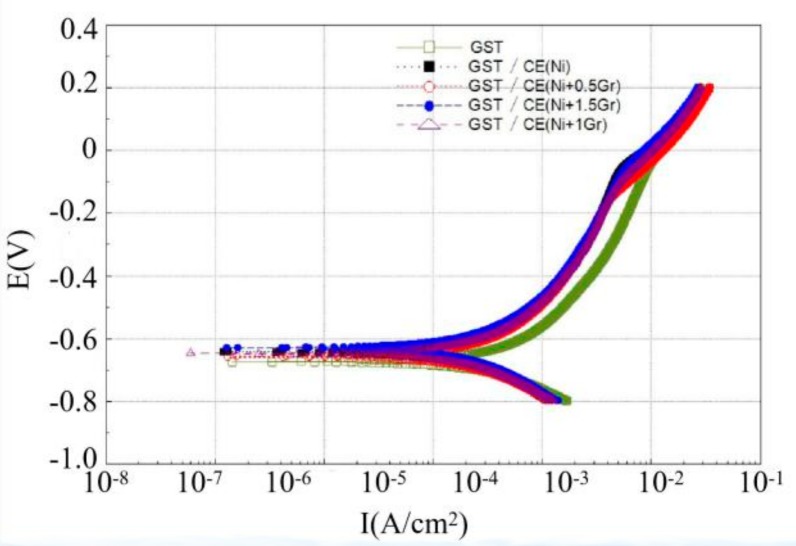
Tafel curve of groove surface texture obtained under different conditions.

**Figure 10 materials-12-03240-f010:**
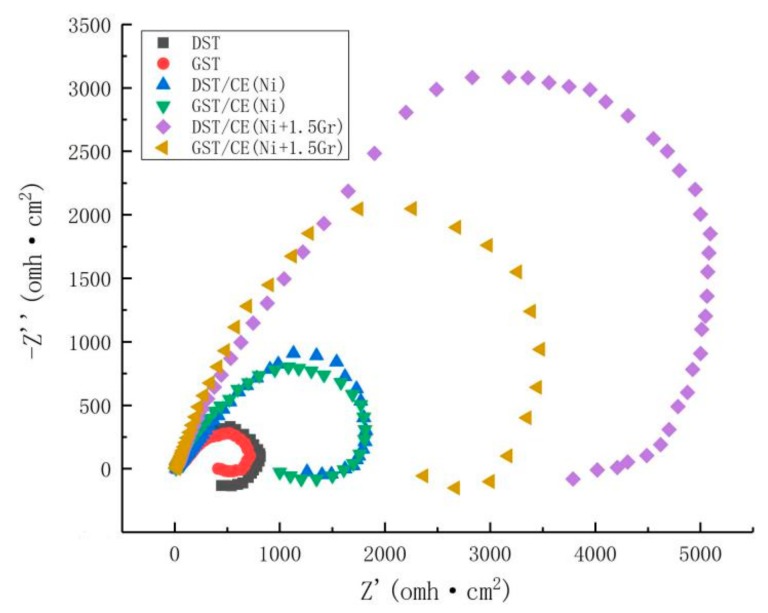
Electrochemical impedance of specimen obtained under different conditions.

**Figure 11 materials-12-03240-f011:**
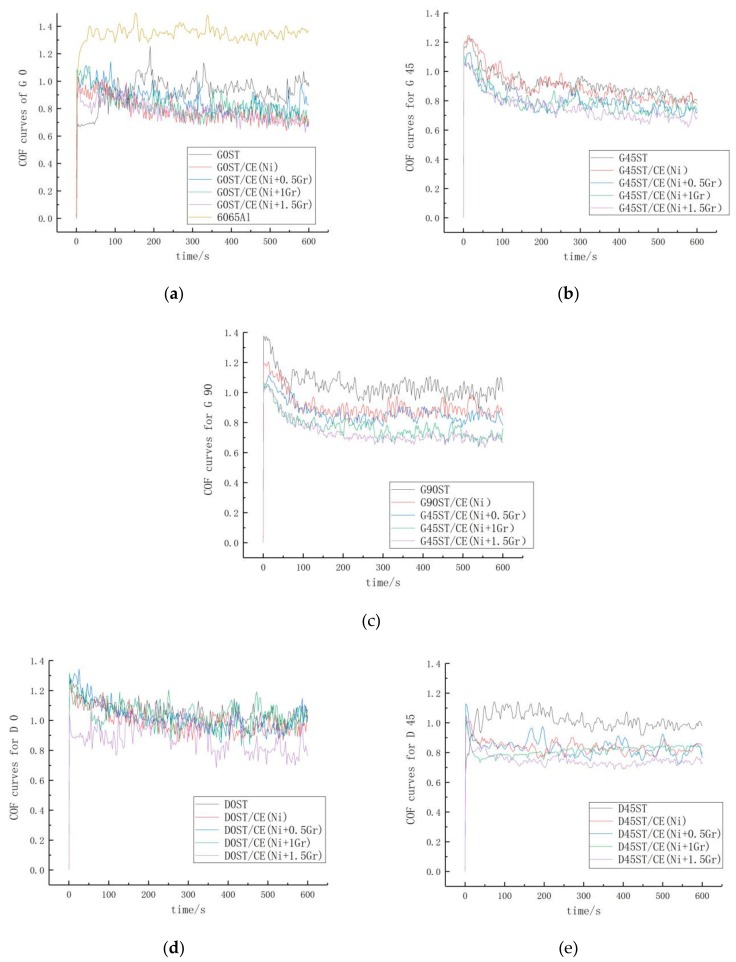
Friction coefficient under stable friction for different ST/CE surface. (**a**–**c**) The stable friction coefficients of GST at 0°, 45° and 90°, respectively. (**d**) and (**e**) the stable friction coefficients of DST at 0° and 45°, respectively.

**Figure 12 materials-12-03240-f012:**
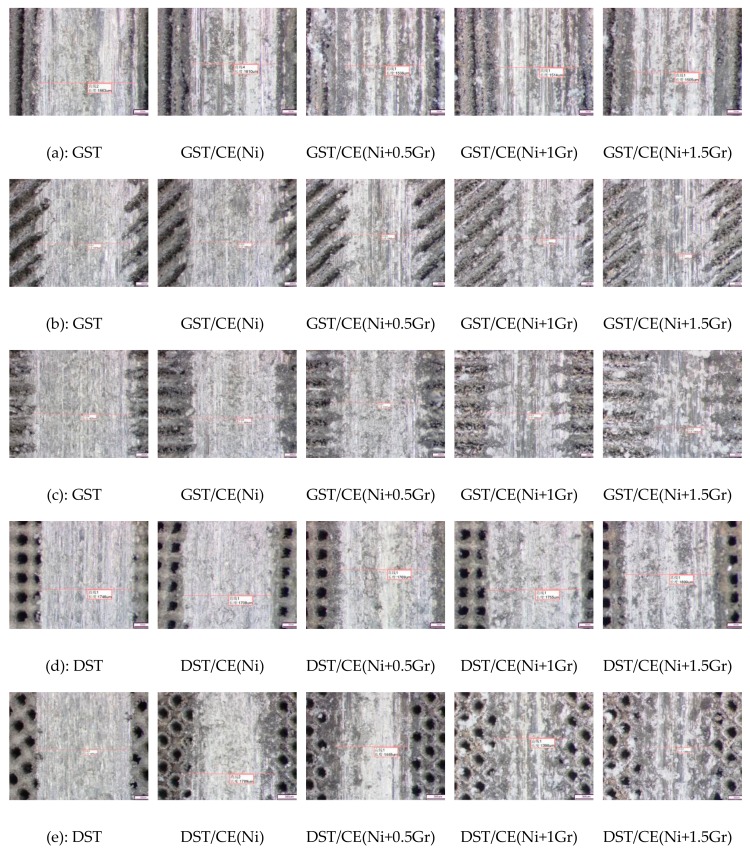
Micrographs of worn trace for all the textured surface of aluminum plate. (**a**–**c**) the worn trace width of GST at 0°, 45° and 90°, respectively. (**d**) and (**e**) the worn trace width of DST at 0° and 45°, respectively.

**Figure 13 materials-12-03240-f013:**
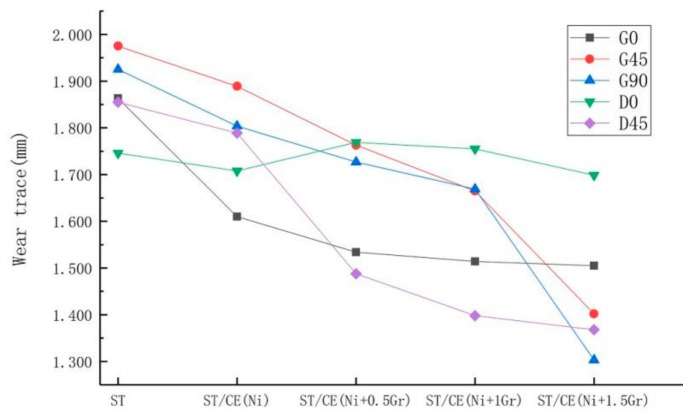
Variation of worn trace width on the surface of different treatment methods.

**Table 1 materials-12-03240-t001:** Nominal chemical compositions of 6065 aluminum alloy (wt %).

Cu	Fe	Mg	Si	Zn	Ti	Bi	Zr	Mn	Al
0.274	0.4	1.028	0.609	0.06	0.06	1.25	0.12	0.05	96.11

**Table 2 materials-12-03240-t002:** Parameters of dimple and groove texture. DST: dimple surface texture; GST: groove surface texture.

Shape	Size of Shape (mm)	Distribution (μm)	Sketch
Diameter	Striate Width	Gap
DST	20 × 20	200	—	200	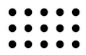
GST	—	200	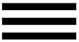

**Table 3 materials-12-03240-t003:** Electroplating conditions.

Component of the electrolyte	Nickel ammonium sulfamate tetrahydrate: 350 g/L
	Nickel chloride: 10 g/L
	Boric acid: 20 g/L
	Citric acid: 5 g/L
	Sodium dodecyl benzene sulfonate: 0.015 g/L
	Deionized water: 450 ml g/L
Electroplating processing parameter	Current density: 13.75 A/dm2
	Electroplating time: 5 min
	Electroplating temperature: 40 °C
	Stirring speed: 300 rpm
	Drying temperature: 200 °C; Drying time: 0.5 min

**Table 4 materials-12-03240-t004:** Corrosion parameters for dimple and groove surface texture base under different conditions.

Sample	E_corr_/mV	I_corr_/(μA/cm^2^)	Sample	E_corr_/mV	I_corr_/(μA/cm^2^)
DST	−655.44	469.92	GST	−675.48	688.7
DST/CE(Ni)	−618.63	468.68	GST/CE(Ni)	−643.1	397.11
DST/CE(Ni+0.5Gr)	−634.04	457.39	GST/CE(Ni+0.5Gr)	−659.69	448.98
DST/CE(Ni+1Gr)	−629.84	415.84	GST/CE(Ni+1Gr)	−649.11	407.4
DST/CE(Ni+1.5Gr)	−601.69	397.18	GST/CE(Ni+1.5Gr)	−629.69	404.58

**Table 5 materials-12-03240-t005:** Friction coefficient of stable friction for all the textured composite electroplating coatings.

Sample	G0	G45	G90	D0	D45
ST	0.889	0.909	1.049	1.045	1.012
ST/CE(Ni)	0.789	0.895	0.907	0.989	0.8358
ST/CE(Ni+0.5Gr)	0.864	0.799	0.865	1.039	0.8323
ST/CE(Ni+1Gr)	0.824	0.797	0.786	1.019	0.8143
ST/CE(Ni+1.5Gr)	0.777	0.755	0.735	0.892	0.7492

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
