# Peer review of "Preparation and Properties of Graphene/Nickel Composite Coating Based on Textured Surface of Aluminum Alloy"

_materials, 2019, doi:10.3390/ma12193240_

Round 1

Reviewer 1 Report

Ref: materials-582109

Title: Preparation and Properties of Graphene/Nickel Composite Coating Based on Textured Surface of Aluminum Alloy

In this paper, preparation and properties of Nickel/Graphene composite coating on textured surface of aluminum alloy were investigated. Detailed analysis using electrochemical, Friction and wear tests was done. However, some points must be addressed before acceptation for publication. So, this reviewer recommends the publication of the present manuscript in "Materials" after major modifications:

Quality of Figures should be improved. OCP plots should be provided in the manuscript along with its discussion. Electrochemical impedance spectroscopy is necessary to characterize the electrochemical behavior and should be performed.

Author Response

Materials-582109

Title: Preparation and Properties of Graphene/Nickel Composite Coating Based on Textured Surface of Aluminum Alloy

Thank you for giving us the opportunity to revise our manuscript and for the thoughtful and valuable comments on this manuscript. We have checked through the whole paper and corrected errors. At same time, we add EDS, SEM and Electrochemical impedance data in the revised paper. All the changes are highlighted in the Marked Revision. The point-to-point responses to the comments are listed as following.

Point 1: Quality of Figures should be improved. OCP plots should be provided in the manuscript along with its discussion

Response 1: In the revised manuscript, we have replaced those figures with high quality. At same time, all figures will be uploaded as attachments for this paper.

Point 2: Electrochemical impedance spectroscopy is necessary to characterize the electrochemical behavior and should be performed. 

Response 2: In the revised manuscript (p.9 lines 278-288), we have added Electrochemical impedance spectroscopy shown as Figure 9and following explanation on it.

Figure 9. Electrochemical impedance of specimen obtained under different conditions

The corrosion resistant properties of textured graphene/nickel composite electroplating coatings were evaluated by the electrochemical impedance tests. The Nyquist plots of all the surfaces with different treatments were shown by Figure 9. Usually, the larger diameter of capacitive loop in Nyquist plot implies higher corrosion resistance [28]. From Figure 9, we can find that DST/CE (Ni+Gr) coatings exhibit remarkable higher impedance compared to GST/CE (Ni+Gr) coatings and the impedance of coating increased with higher graphene content. These results are consistent with that of potentiometric polarization curve analysis. DST/CE (Ni+Gr) coating have the highest impedance and better corrosion resistance in plating bath containing the largest amount of graphene addition.

Reviewer 2 Report

In this manuscript the authors evaluated the effect of a composite coating, namely made of graphene and nickel, on a textured surface of an Al-Si-Mg alloy (6065) by measuring the corrosion and wear resistance, considering also how the percent of graphene could influence the system properties. The present study is within the scope of the journal, could be of a particular scientific interest and is quite innovative. The manuscript is well written in terms of English language but misses some major aspects. However, in my opinion this is not sufficient to be published as a scientific article. The results and discussion sections appear to me weak in some parts, with not enough elements to justify some statements. In particular, chemical aspects related to the properties of the coating should be better defined. Is the coating well adherent to the alloy? In what way? How the authors can justify the homogeneity of the coating? if the coating applied is uniform is it possible that it is also multi- layered? How the graphene and the Nickel atoms are able to interact with each other and how this affect the nature of the coating in terms of physical and chemical properties?

In conclusion, the content of this study is not sufficient for a paper which can be published in Materials but could be resubmitted after a thorough revision. I suggest that the authors should focus more on the surface characterization of the coating before testing it. This should help in the understanding the behavior of the system according to the different parameters chosen.

Specific comments

Abstract

When the authors state:”..can improve the electrical corrosion parameters…” they should consider that corrosion parameters can be kinetic or thermodynamic and that the term electrical is not enough specific.

Introduction

To my opinion it is not very clear the purpose of the study and the choice of an Al-Si-Mg alloy. The authors should describe more the aim of the study and should define the specific filed of applicability of the chosen materials. In my opinion it is also important to stress the purpose and future impact of the study of future researches.

Page 1 line 34: maybe it would be helpful if the authors give the exact name of the material (CNT-carbon nanotubes, MoS2-…) and not only the acronym.

Also SI should be followed: V for volts; L for liters, rpm and not r/min, pH and not PH

Page 2 line 61. Reference should be added at the end of the sentence.

Experimental procedure

In the experimental session I would suggest describing the procedure more thoroughly, maybe referring more to figure 1. Furthermore, some details are missing, for instance the polishing procedure is not well specified, the different conditions of the experiments are just hinted and not defined (a table would help in understanding the different sub-systems defined). This is very important because the experiment could be reproduced or replicated by other authors.

Did the authors perform the analysis on replicated samples? If they did, they should explicit the number of replicates in order to have a good data dispersion, a statistical validation and a significance of the measurements.

To me it is important to add a section that should envisage the use of Scanning Electron microscopy SEM or Atomic force microscopy AFM to help characterize the coatings before the tests.

2.2 Surface texture construction: did the authors consider the introduction of defects provided by plasma?

2.4.2 Corrosion resistance: did the authors perform potentiodynamic polarization after the reach of the equilibrium of the system? Did they wait long enough to achieve the formation of the Al2O3 oxide in the surface of the “nude” material?

Can the authors explain the use of 1 mV/s for the scanning rate Is the scan rate adequately low in order to have an good response from the system?

Results and discussion

In the “morphologies of the coatings and textured surface” section it is important to me have some detailed information on the coatings in terms of visual inspection. SEM and AFM could help in the description of the morphology and in the identification of the micro and nano characteristics and variations of the systems.

Page 6 line 202: “These dimensions variations…continuous coating” how can the authors justify the sentence?

Corrosion resistant properties of the electroplating coating:

Authors should be more careful when describe the attribution of system behavior to a certain phenomenon without post experiment analyses. They could help to strongly support their thesis. Furthermore, did the author consider the fact that aluminum alloys are passive materials and the possibility of localized corrosion has more chances to occur than general corrosion? Some results are not mentioned and Samples GST show a different behavior comparing to those DST, since it seems that corrosion current densities are the lowest for the pure Ni coating. Can the author explain it? Was it connected to the surface characteristics or to the change in the coating properties?

A more thorough discussion section should be added because in this form the article looks more like a technical report than a scientific article.

Figures:

Figure 1 is not well described in the text and this would help in the understanding of the methodological procedure.

Figure 3 and 4: the labels and the markers are not visible and should be bigger.

Figure 6. Current density is A/cm2 or A*cm-2

Author Response

Point-to-point responses to the comments

Materials-582109

Title: Preparation and Properties of Graphene/Nickel Composite Coating Based on Textured Surface of Aluminum Alloy

Reviewer 2

 Thank you for giving us the opportunity to revise our manuscript and for the thoughtful and valuable comments on this manuscript. We have checked through the whole paper and corrected errors. At same time, we add EDS, SEM and Electrochemical impedance data in the revised paper. All the changes are highlighted in the Marked Revision. The point-to-point responses to the comments are listed as following. Please find Marked version of this paper in the attachment.

Point 1: Chemical aspects related to the properties of the coating should be better defined.

Response 1: In the revised manuscript (p.7 lines 224-228), we have added EDS result and some statements about “Chemical aspects”.

Figure 6  (g) EDS result of graphene/Ni composite electroplating coating

The EDS analyses indicate that graphene were successfully adsorbed on the graphene/Ni composite electroplating coating during electrodeposition, which could greatly increases the number of nucleation sites for Ni2+ to reduce, resulting in a fine grained and intact arrangement of Ni crystals [25]. The chlorine are shown in EDS and it may be introduced by nickel chloride during electroplating.

Point 2: Is the coating well adherent to the alloy? In what way? How the authors can justify the homogeneity of the coating?

Response 2: In the revised manuscript (p.7 lines 210-214), we have added the sectional images of electroplating coatings (as shown in Figure 5) and explanation. 

As can be seen from Figure 5, sectional images of the GST electroplating coatings show the electroplating coatings and the alloy matrix are uniformly and tightly connected. Compared with the textured nickel coating, the GST graphene/Ni composite coating has better bonding with the Al alloy substrate, and there are no obvious impurities and defects at the interface.

Point 3: If the coating applied is uniform is it possible that it is also multi- layered? How the graphene and the Nickel atoms are able to interact with each other and how this affect the nature of the coating in terms of physical and chemical properties?

Response 3: In the revised manuscript (p.7 lines 214-224), we have added the SEM images of composite electroplating coatings (as shown in Figure 6) and some statements.

Figure 6. SEM images of the textured graphene/nickel composite electroplating coatings. (a-c) GST/CE (Ni-1.5Gr). (d-f) DST/CE (Ni-1.5Gr) samples. (g) EDS result of graphene/Ni coating

During the composite electrodeposition, Ni2+s would be preferably electrodeposited onto the graphene and lead to the formation of microbulges on the electroplating coatings. In the process of composite electrodeposition, Ni2+s would be preferentially electrodeposited onto graphene and encapsulated graphene which finally lead to form micro-convex bulges on the electroplating surface. This phenomenon can be found in the Figure6 (c) and Figure 6(f). So the surface morphology of the composite electroplating coating with the different contents of graphene is probably correlated to the electric force between the composite and the Ni matrix during deposition, which influences the absorption of composite on the surface of the electrodepositing layer [24].

Point 4: Abstract: When the authors state:”..can improve the electrical corrosion parameters…” they should consider that corrosion parameters can be kinetic or thermodynamic and that the term electrical is not enough specific.

Response 4: In the revised manuscript (p.1 lines 22-23), we have checked and corrected it in abstract.

Electrochemical corrosion tests results show that the corrosion resistance of textured graphene/Ni composite electroplating coating is related to the specific surface texture, and the dimple texture can improve the electrical corrosion parameters, like electrode potential greatly.

Point 5: Introduction:Page 1 line 34: maybe it would be helpful if the authors give the exact name of the material (CNT-carbon nanotubes, MoS2-…) and not only the acronym.

Response 5: In the revised manuscript (p.1 lines 36-38), we have checked and corrected it in Introduction.

Recent research have found that the addition of materials such as SiC, CNT-carbon nanotubes, nano-diamond and MoS2-nanosheet to the composite plating solution results in a certain improvement in corrosion resistance and friction resistance.

Point 6: Also SI should be followed: V for volts; L for liters, rpm and not r/min, pH and not PH.

Response 6: In the revised manuscript (p.4 lines 149; 158), we have checked and corrected it through the whole paper.

The pH value of the solution obtained was 3-4; and the stirring speed is 300 rpm.

Point 7: Page 2 line 61. Reference should be added at the end of the sentence.

Response 7: In the revised manuscript (p.2 lines 62-63), we have added more references on it.

This is the reason that graphene can prevent the occurrence and development of defect corrosion at these locations in cracks, gaps and micro-pores in the Ni matrix [11].

Point 8: Experimental procedure: In the experimental session I would suggest describing the procedure more thoroughly, maybe referring more to figure 1. Furthermore, some details are missing, for instance the polishing procedure is not well specified, the different conditions of the experiments are just hinted and not defined (a table would help in understanding the different sub-systems defined). This is very important because the experiment could be reproduced or replicated by other authors.

Response 8: In the revised manuscript (p.4 lines 158-160), we have added Table 3 to define experimental procedure.

The samples were then washed with deionized water after electroplating process and finally put into the SX2-4-10 electric furnace (Yahua Co. Ltd. China) at 200°C for 0.5 minutes for drying. And detailed electroplating parameter were shown in Table 3.

Table 3. Electroplating conditions

Component of the electrolyte

Nickel ammonium sulfamate tetrahydrate: 350 g/L

Nickel chloride: 10 g/L

Boric acid: 20 g/L

Citric acid: 5 g/L

Sodium dodecyl benzene sulfonate: 0.015 g/L

Deionized water: 450ml g/L

processing parameter

Electroplating time: 5 min

Electroplating temperature: 40°C

Stirring speed: 300 rpm

Drying temperature : 200°C; Drying time: 0.5 min

Point 9: Did the authors perform the analysis on replicated samples? If they did, they should explicit the number of replicates in order to have a good data dispersion, a statistical validation and a significance of the measurements.

Response 9: In the revised manuscript (p.7 lines 208-210), we have added some statements about it.

It is found that the pit diameter and groove width of the samples respectively are in good agreement by measuring several different positions on the surface of the sample, which indicates that the coating is continuous and uniform.

Point 10: To me it is important to add a section that should envisage the use of Scanning Electron microscopy SEM or Atomic force microscopy AFM to help characterize the coatings before the tests.

Response 10: In the revised manuscript (p.7 lines 214-224), we have added the SEM images of electroplating coatings and some statements about the result which can be found in Response 3.

Figure 6. SEM images of the textured graphene/nickel composite electroplating coatings. (a-c) GST/CE (Ni-1.5Gr). (d-f) DST/CE (Ni-1.5Gr) samples. (g) EDS result of graphene/Ni coating.

Point 11:  2.2 Surface texture construction: did the authors consider the introduction of defects provided by plasma?

Response 11: In the experimental feasibility stage, it takes so long to determine many experimental parameters, so the introduction of defects provided by plasma in electroplating have been minimized.

Point 12: 2.4.2 Corrosion resistance: did the authors perform potentiodynamic polarization after the reach of the equilibrium of the system? Did they wait long enough to achieve the formation of the Al2O3 oxide in the surface of the “nude” material? Can the authors explain the use of 1 mV/s for the scanning rate is the scan rate adequately low in order to have a good response from the system?

Response 12: The results of electrochemical corrosion experiments are obtained after the system equilibrium and are carried out in accordance with standard electroplate procedures. In the revised manuscript (p.5 lines 175-177), we have added the explanation has been added to corresponding part.

Tafel lines were recorded at a scanning rate of 1 mV/s, which is low enough to have a good response for the system, from 0.8-0.2V versus open circuit potential.

Point 13:  Results and discussion: In the ”morphologies of the coatings and textured surface” section it is important to me have some detailed information on the coatings in terms of visual inspection. SEM and AFM could help in the description of the morphology and in the identification of the micro and nano characteristics and variations of the systems. 

Page 6 line 202: “These dimensions variations…continuous coating” how can the authors justify the sentence?

Response 13: In the revised manuscript (p.7 lines 210-214), we have added the sectional images of electroplating coatings and the SEM images of electroplating coatings. Here, same response as point 2.

Figure 5. Sectional images of the textured electroplating coatings. (a) Composite electroplating coatings(Ni)  (b) Composite electroplating coatings(Ni-1.5Gr)

Figure 6. SEM images of the textured graphene/nickel composite electroplating coatings. (a-c) GST/CE (Ni-1.5Gr). (d-f) DST/CE (Ni-1.5Gr) samples. (g) EDS result of graphene/Ni coating.

Point 14: Corrosion resistant properties of the electroplating coating: Authors should be more careful when describe the attribution of system behavior to a certain phenomenon without post experiment analyses. They could help to strongly support their thesis. Furthermore, did the author consider the fact that aluminum alloys are passive materials and the possibility of localized corrosion has more chances to occur than general corrosion? Some results are not mentioned and Samples GST show a different behavior comparing to those DST, since it seems that corrosion current densities are the lowest for the pure Ni coating. Can the author explain it? Was it connected to the surface characteristics or to the change in the coating properties?

Response 14: Since the experimental area of the electrochemical corrosion test in this paper is 1 cm 2 (this is the requirement of the standard sample) and the area is small, the distinction between "localized corrosion" and "general corrosion" is not significant. In the revised manuscript (p.9 lines 275-277), we have added the explanation of “corrosion current densities are the lowest for the pure Ni coating”

The corrosion current density of GST/CE (Ni) is less than that of DST/CE (Ni+1.5Gr) because the two values are too close, so the corrosion resistance of dimple textured composite coatings is more explained by the comparison of corrosion potential.

Point 15: Figures: Figure 1 is not well described in the text and this would help in the understanding of the methodological procedure. Figure 3 and 4: the labels and the markers are not visible and should be bigger. Figure 6. Current density is A/cm2 or A*cm-2

Response 15: In the revised manuscript (p.3 lines118-120), we have changed those figures into high quality. All figures will upload to Materials by attachment.

Here, Texture surface was fabricated by using laser processing and then the graphene/Ni composite electroplating processing was carried out on the textured surface.

 Reviewer 3 Report

 Preparation and Properties of Graphene/Nickel Composite Coating Based on Textured Surface of Aluminum Alloy by Xu et al.

This is an experimental work on fabricating graphene/nickel composite on textured aluminum alloy. The authors state that this material has best wear properties under vertical friction. This work is of interest to material science and aluminum-alloy based research. However, the paper is poorly written and extensive English editing is required before is published. Therefore, a major revision is needed. The authors need to extensively revise their text and follow the suggestions below:

Labels on figures 3, 4, and 8 are unreadable and need to be revised. Figure 7 needs to be at a higher resolution since lines are washed-out. In the abstract the sentence “ Striate and pitted textured surfaces were fabricated by laser with 6065 aluminum alloy (6065Al) as matrix material, and nickel/graphene (Ni/Gr) composite electroplating was carried  out on the textured surface, and graphene / nickel composite electroplating coating was prepared  on the textured surface of 6065 aluminum alloy.”  is too long and needs to be fragmented.  Similar run-on sentences appear throughout the text. For example, on page 9 “From Figure 7 and Table 4, it can be seen that under the same friction conditions, the friction coefficients of ST are larger than those of ST/CE(Ni) which shows that the wear resistance of nickel-based coatings is obviously improved after pure nickel plating, while the friction coefficient of textured graphene/nickel composite coatings decreases gradually with the increase of graphene content.” needs to be fragmented. The introduction is too long and need to be shortened. In conclusions 1) and 2) should be merged and shortened.

Author Response

Point-to-point responses to the comments

 Materials-582109

Title: Preparation and Properties of Graphene/Nickel Composite Coating Based on Textured Surface of Aluminum Alloy

Thank you for giving us the opportunity to revise our manuscript and for the thoughtful and valuable comments on this manuscript. We have checked through the whole paper and corrected errors. At same time, we add EDS, SEM and Electrochemical impedance data in the revised paper. All the changes are highlighted in the Marked Revision. The point-to-point responses to the comments are listed as following. Please find Marked version of this paper in the attachment.

Point 1: Labels on figures 3, 4, and 8 are unreadable and need to be revised. Figure 7 needs to be at a higher resolution since lines are washed-out.

Response 1: In the revised manuscript, we have changed those figures into high quality. All figures will be uploaded to Materials by attachment.

Point 2: In the abstract the sentence”Striate and pitted textured surfaces were fabricated by laser with 6065 aluminum alloy (6065Al) as matrix material, and nickel/graphene (Ni/Gr) composite electroplating was carried  out on the textured surface, and graphene / nickel composite electroplating coating was prepared  on the textured surface of 6065 aluminum alloy.”  is too long and needs to be fragmented.

Response 2: In the revised manuscript (p.1 lines 12-16), we have checked and rewritten this sentence. 

Regular arrayed dimple surface texture (DST) and groove surface texture (GST) were fabricated by using laser processing on 6065 aluminum alloy matrix (6065Al), then the electrodeposition of Ni and Gr/Ni coatings on textured surface of aluminum alloy base was then performed in electrolytes with concentrations of 0、0.5、1 and1.5mg graphene.

Point 3: For example, on page 9 “From Figure 7 and Table 4, it can be seen that under the same friction conditions, the friction coefficients of ST are larger than those of ST/CE(Ni) which shows that the wear resistance of nickel-based coatings is obviously improved after pure nickel plating, while the friction coefficient of textured graphene/nickel composite coatings decreases gradually with the increase of graphene content.” needs to be fragmented.

Response 3: In the revised manuscript (p.11 lines 298-301), we have checked and corrected this sentence.

From Figure 10 and Table 5, it reveals that the friction coefficients of ST are higher than those of ST/CE(Ni) under same friction conditions, which shows that the wear resistance is obviously improved after pure nickel plating. At same time, the friction coefficient of textured graphene/Ni composite coatings decreases gradually with the increase of graphene content.

Point 4: The introduction is too long and need to be shortened. In conclusions 1) and 2) should be merged and shortened.

Response 4: In the revised manuscript (p.13 lines 358-368), we have checked and rewrite the introduction and conclusion.

The following conclusions can be drawn from the present study:

(1).Graphene/Ni composite electroplating coating on groove and dimple textured 6065Al surface were prepared in this paper. Sectional images of the samples show the electroplating coatings and alloy matrix are uniformly and tightly connected uniformly and tightly. Electrochemical corrosion test results show that the corrosion resistance of the dimple textured graphene/Ni composite electroplating coating is better than that of the groove textured one due to the effect of those small air pockets created by dimple textured surface.

(2).Friction and wear experiments revealed that the surface friction properties are related to the texture, the amount of graphene added and the direction of friction. Anti-friction performance of the groove textured graphene/Ni composite electroplating coating is higher than that of the dimple textured one.  

Reviewer 4 Report

The work discussed the subject of the Preparation and Properties of Graphene/Nickel Composite Coating Based on Textured Surface of Aluminum Alloy. Presented in the manuscript research shows a low level of significance at the present form and from the above reason and taking care of the high level of research target to Materials journal, I recommend rejecting or major revision of this article.

The general problem of the work is the attempt to sell some research results without the major material background and it characteristic, that remains the main scope of the journal and that is tightly hidden in the manuscript methodological subsection. The authors do not characterize the samples to confirm the composite character of the modified surface. Also, a numerous technological operation during final sample preparation remains only steps without proper explanation and justification like for example finally drying in 200°C for 0.5 minutes (p.4 l.138-162)? No literature reference for the electrodeposition process suggests also that procedure remain the author's original achievement without however proper justification and the way of its deduction for finally obtained results.

After laser treatment not only the surface roughness change the authors doesn't mention anything about the process atmosphere and possible surface reaction of the substrate this indeed change the view of analysed relation and further reaction. No data confirm the substrate composition and its possible changes after laser treatment. The microscopic view shows only prepared growes or dimples texture. The scale bares and included information on the figures 3,4,8 remain not visible. Obtained corrosion resistance and wear tests results shows slight differences clearer visible for friction coefficients, still without base material characteristic stays unuseful and speculative for any further conclusions.

Other remarks

For the substrate material, the state of delivery should be known.

In the abstract, the expression matrix material remains not properly used, the aluminium alloy remains a substrate material. (p.1 l.14)

Author Response

Materials-582109

Title: Preparation and Properties of Graphene/Nickel Composite Coating Based on Textured Surface of Aluminum Alloy

Thank you for giving us the opportunity to revise our manuscript and for the thoughtful and valuable comments on this manuscript. We have checked through the whole paper and corrected errors. At same time, we add EDS, SEM and Electrochemical impedance data in the revised paper. All the changes are highlighted in the Marked Revision. The point-to-point responses to the comments are listed as following. Please find Marked version of this paper in the attachment.

Point 1: a numerous technological operation during final sample preparation remains only steps without proper explanation and justification like for example finally drying in 200°C for 0.5 minutes (p.4 l.138-162)?

Response 1: In the revised manuscript (p.4 lines 158-160), we have added this part into requirement, and have added Table 4 on Experimental procedure for detailed explanation.

The samples were then washed with deionized water after electroplating process and finally put into the SX2-4-10 electric furnace (Yahua Co. Ltd. China) at 200℃ for 0.5 minutes for drying. 

Table 3. Electroplating conditions

Component of the electrolyte

Nickel ammonium sulfamate tetrahydrate: 350 g/L

Nickel chloride: 10 g/L

Boric acid: 20 g/L

Citric acid: 5 g/L

Sodium dodecyl benzene sulfonate: 0.015 g/L

Deionized water: 450ml g/L

Electroplating processing parameter

Current density: 13.75 A/dm2

Electroplating time: 5 min

Electroplating temperature: 40°C

Stirring speed: 300 rpm

Drying temperature : 200°C; Drying time: 0.5 min

Point 2: No literature reference for the electrodeposition process suggests also that procedure remain the author's original achievement without however proper justification and the way of its deduction for finally obtained results.

Response 2: In the revised manuscript (p.4 lines 145-149), we have added the references.

Electrolyte[23], as can be seen in Table 3, was prepared by adding nickel ammonium sulfamate tetrahydrate (350 g/L), nickel chloride (10 g/L), boric acid (20 g/L), citric acid (5 g/L) and sodium dodecyl benzene sulfonate (0.015 g/L) into 450 ml deionized water and then stirred by JJ-1 stirrer (XinRui Co, Ltd. China) for 10 minutes in a water bath at 40℃.

Point 3: The microscopic view shows only prepared growes or dimples texture. The scale bares and included information on the figures 3,4,8 remain not visible.

Response 3: In the revised manuscript, we have changed those figures into high quality. All figures will be uploaded to Materials by attachment.

Round 2

Reviewer 1 Report

Manuscript can be accepted for publication.

Author Response

Thanks very much for your kind work and consideration on publication of our paper. On behalf of my co-authors, we would like to express our great appreciation for your thoughtful and valuable comments on this manuscript.

Reviewer 2 Report

The authors have responded thoroughly to my remarks and made changes that in my opinion are relevant. Therefore I recommend the paper for publication.

Author Response

(The authors gave the same response as above.)

Reviewer 3 Report

The paper has been substantially revised and should be accepted in the present form.

Author Response

(The authors gave the same response as above.)

Reviewer 4 Report

The work discussed the subject of the Preparation and Properties of Graphene/Nickel Composite Coating Based on Textured Surface of Aluminum Alloy. Presented in the manuscript research shows a low level of significance at the present form and from the above reason and taking care of the high level of research target to Materials journal, I recommend rejecting or major revision of this article.

The general problem of the work remain unchanged a major material background and it's characteristic, are not present in the manuscript. The authors do not characterize the samples to confirm the composite character of the modified surface. Additional SEM and EDS investigation don’t bring anything new also some discussion stays speculative in reference to new data. A numerous technological operation during final sample preparation remains only steps without proper explanation and justification. No literature reference for the electrodeposition process suggests also that procedure remain the author's original achievement without however proper justification and the way of its deduction for finally obtained results.

After laser treatment not only the surface roughness change the authors doesn't mention anything about the process atmosphere and possible surface reaction of the substrate this indeed change the view of analysed relation and further reaction. No data confirm the substrate composition and its possible changes after laser treatment. The microscopic view shows only prepared growes or dimples texture. Obtained corrosion resistance and wear tests results shows slight differences clearer visible for friction coefficients, still without base material characteristic stays unuseful and speculative for any further conclusions.

Author Response

The authors would like to appreciate you for thoughtful and valuable comments on this manuscript. We have studied your comments carefully and have made revision which marked in the paper. We have tried our best to revise our manuscript according to the comments. The point-to-point responses to the comments are listed as following.

Point 1: The general problem of the work remain unchanged a major material background and it's characteristic, are not present in the manuscript.

Response 1: In the revised manuscript (p.1 lines 31-33, p.3 lines 118), we have added a major material background and it's characteristic in Introduction.

Aluminum alloy has been considered a good choice as a material in the aviation, aerospace, automotive, marine and chemical industries, due to its excellent specific strength and plasticity, high electrical conductivity, thermal conductivity and remarkable corrosion resistance [1].

6065Al (Density 2700kg/m3, Elastic modulus 67Gpa and Yield strength 139Mpa)

Point 2: The authors do not characterize the samples to confirm the composite character of the modified surface. Additional SEM and EDS investigation don’t bring anything new also some discussion stays speculative in reference to new data.  

Response 2: In the revised manuscript (p.8 lines 224), we have added the confirmation of related SEM and EDS investigation by similar observation shown in another article. On the other hand, since the scanning speed of laser processing is too fast (200 mm/s) for treated material to react in the air, in this case, the surface reaction and substrate composition changes of the substrate can be neglected.

and smilar observation was reported by Jiang [10].

Point 3: A numerous technological operation during final sample preparation remains only steps without proper explanation and justification. No literature reference for the electrodeposition process suggests also that procedure remain the author's original achievement without however proper justification and the way of its deduction for finally obtained results.

Response 3: In the revised manuscript (p.5 lines 181-182; p.9 lines 261-263), we have added the literature reference for the electrodeposition process.

Electrochemical impedances were obtained in the frequency range of 0.01 Hz–10 kHz and the amplitude of sinusoidal signal was 10 mV[24]. These tests were carried out in neutral 3.5% NaCl solution at room temperature[25].

This result is consistent with another study [10], in which, more graphene added play a much more obvious role in improving the corrosion resistance during the electrodeposition of Ni.

Point 4: After laser treatment not only the surface roughness change the authors doesn't mention anything about the process atmosphere and possible surface reaction of the substrate this indeed change the view of analysed relation and further reaction. No data confirm the substrate composition and its possible changes after laser treatment.

Response 4: 

Laser processing is a common processing method in modern processing. Generally, it can be carried out at room temperature without adding auxiliary measures such as gas atmosphere protection.

Moreover, since the scanning speed of laser processing is too fast (200 mm/s) for treated material to react in the air, in this case, the surface reaction and substrate composition changes of the substrate can be neglected.

On the other hand, it is true that laser processing can improve the hardness of substrate which is also benefit for the enhancement of properties of textured substrate. Acutually, we have tested micro-hardness of all samples by using HV-1000A micro-hardness tester and hardness value of aluminum alloy substrate and laser textured substrate are shown in the following Figures 1 and 2. As shown in Figure 1, the hardness indentation of the aluminum alloy matrix can be clearly observed, but it is difficult to observe the specific hardness indentation on the textured surface, which can only be estimated roughly.

Indeed it would be valuable to compare hardness between substrate and textured substrate, but it is actually very difficult to obtain clear hardness indentation on the textured surface and also considering the manuscript concise, we did not add these figures in the revised manuscript.

In addition, the purpose of laser processing here is to obtain texture surfaces with different morphologies (GST/DST). Therefore, the effect of laser on micro-hardness has not been discussed too much in this paper.

Figure1. Hardness Indentation on the surface of different samples.

Figure2. Micro-Hardness for Different Samples.

Point 5: The microscopic view shows only prepared growes or dimples texture. Obtained corrosion resistance and wear tests results shows slight differences clearer visible for friction coefficients, still without base material characteristic stays unuseful and speculative for any further conclusions.

Response 5: The friction and wear experiments of untreated 6065 aluminum alloy substrate have been carried out in our earlier experimental stage. As shown in Figure 3, the friction test results show that the real-time friction coefficient of untreated substrate varies sharply, and the stability value is around 1.3, which is much higher than that of textured composite graphene electroplated substrate. Therefore, the textured surface of aluminum alloy electroplated with composite graphene has much better properties. Since this paper focus on the influence of texture and graphene on the properties of substrate, we did not add this result and figure into the revised manuscript.

It is true the obtained corrosion resistance and wear tests results shows slight differences clearer visible for friction coefficients between different textured substrate. However, from the electrochemical corrosion point of view, the distinction is relatively obvious. This result provides us and related researchers information useful in the future investigations.

Figure3. Friction of Coefficient under stable Friction for 6065Al.

Round 3

Reviewer 4 Report

The work discussed the subject of the Preparation and Properties of Graphene/Nickel Composite Coating Based on Textured Surface of Aluminum Alloy. Presented in the manuscript research shows a low level of significance at the present form and from the above reason and taking care of the high level of research target to Materials journal, I recommend rejecting or major revision of this article.

The general problem of the work remains unchanged, a major material background and it's characteristic understand as research that confirms material composition and structure is not present in the manuscript. The authors characterize final properties without proper judgment of material state and structure. The authors do not characterize the samples to confirm the composite character of the modified surface. Additional SEM and EDS investigation don’t bring anything new, also some discussion stays speculative in reference to new data. No data confirm the substrate composition and its possible changes after laser treatment. Obtained corrosion resistance and wear tests results shows slight differences clearer visible for friction coefficients, still without base material characteristic stays unuseful and speculative for any further conclusions.

Author Response

The authors would like to appreciate you for thoughtful and valuable comments on this manuscript. We have studied your comments carefully and have made revision which marked in the paper. We have tried our best to revise our manuscript according to the comments. The point-to-point responses to the comments are listed as following.

Point 1: 

The general problem of the work remains unchanged, a major material background and it's characteristic understand as research that confirms material composition and structure is not present in the manuscript.  

Response 1: In the revised manuscript (p.3 lines 124-126), we have added a major material background and it's characteristic in Table 1.

The material selected for the present research work is Aluminum 6065 alloy which is commercially available with dimension of 70 mm×20 mm×2 mm under cold-rolled condition and its composition is given in Table 1. Surface of 6065Al plate were ground by grit papers for 10 minutes in order to remove oxide and obtain a smooth surface.

Table 1. Compositions of 6065 aluminum alloy (wt %).

Cu

Fe

Mg

Si

Zn

Ti

Bi

Zr

Mn

Al

0.274

0.4

1.028

0.609

0.06

0.06

1.25

0.12

0.05

96.11

Point 2: 

The authors characterize final properties without proper judgment of material state and structure. The authors do not characterize the samples to confirm the composite character of the modified surface.

Response 2: In the revised manuscript, the authors compare the morphology before and after electroplating by using the width of groove, diameter of dimple, section images of coatings and SEM images of the surface microbulges (Figure 6). Macro final properties are characterized by using wear resistance and corrosion resistance. The composite character of the modified surface can be defined by the SEM images and the results of EDS. In the revised manuscript (p.8 lines 228-231), we have added more about description about EDS result.

The EDS result in Figure 7(g) gives that the content of C and Ni are estimated to be 3.496% and 64.214% respectively, which indicates that graphene were successfully adsorbed on the graphene/Ni composite electroplating coating during electrodeposition.

Point 3: 

Additional SEM and EDS investigation don’t bring anything new, also some discussion stays speculative in reference to new data.

Response 3: In the revised manuscript (p.8 lines 224-225, p.8 lines 228-231), we have added more about description about EDS result and more reference to confirm SEM images.

This phenomenon can be found in the Figure 6(c) and Figure 7(f) and similar observation was also reported by Jiang [10].

The EDS result in Figure 7(g) gives that the content of C and Ni are estimated to be 3.496% and 64.214% respectively, which indicates that graphene were successfully adsorbed on the graphene/Ni composite electroplating coating during electrodeposition.

Point 4: 

No data confirm the substrate composition and its possible changes after laser treatment. 

Response 4: In the revised manuscript (p.4 lines 135-137), we have added content of the hardness change and related Figure on substrate after laser treatment.

After laser treatment, the hardness of substrate (41.4 HV) can be improved to 102.2 HV after laser processing, which is benefit for the enhancement of properties of textured substrate (as shown in Figure 2).

 Figure 2. Micro-hardness Indentation on the surface of different samples.

As for substrate composition change after laser treatment, we did not mention in the article. It is true that the chemical composition and micro-structure of the material will change to some extent after laser heat treatment. However, since the rapid scanning speed of the laser, the change of chemical composition on the performance is not very obvious. The improvement of hardness mainly comes from micro-structure change brought by the rapid heating and cooling speed of the laser.

Since the purpose of laser processing here is to obtain texture surfaces with different morphologies (GST/DST) and the focus of this paper is on the effect of different textures on the properties under the same laser processing parameter, so there is no much analysis on the chemical composition changes after laser texture. Similar analysis on laser treatment can be found in other references [1-3].

At same time, this is a good suggestion and we will supplement this in the following-up research work.

[1]Meng, C.; Zhou, H.; Cong D.L.; Wang, C.W.; Zhang P.; Zhang Z.H.; Ren, L.Q. Effect of biomimetic non-smooth unit morphology on thermal fatigue behavior of H13 hot-work tool steel. Optics and Laser Technology 2012, 44, 850-859.

[2]Bonse, J.; Koter, R.; Hartelt, M.; Spaltmann, D.; Pentzien, S.; Hohm, S.; Rosenfeld, A.; Krüger, J. Femtosecond laser-induced periodic surface structures on steel and titanium alloy for tribological applications. Applied Physics A 2014, 117, 103-110.

[3]Zhang Z.H.; Shao, F.X.; Liang Y.H.; Lin, P.Y.; Tong, X.; Ren, L.Q. Wear Behavior of Medium Carbon Steel with Biomimetic Surface Under Starved Lubricated Conditions. Journal of Materials Engineering and Performance 2017, 7, 1-11.

Point 5: 

Obtained corrosion resistance and wear tests results shows slight differences clearer visible for friction coefficients, still without base material characteristic stays unuseful and speculative for any further conclusions.

Response 5: In the revised manuscript (p.11 lines 306-312), the friction coefficient curve of base material is added in Figure 11 in our revised manuscript for comparison and corresponding expiation have been added in the related sentences.

It is true the obtained tests results shows slight differences clearer visible for friction coefficients between different textured substrate with ST/CE (Ni-Gr). But there are still much higher than those of ST/CE (Ni) and untreated base material 6065Al (1.353). On the other hand, from the electrochemical resistance point of view, the distinction of electrochemical impedance is relatively obvious. Therefore, this research result may be useful in improving the corrosion resistance of materials in the future. At same time, we will improve the plating process and texture morphology to obtain higher wear resistance of material in future research.

Figure 11. Friction coefficient under stable friction for different ST/CE surface and untreated base material.

From Figure 11 and Table 5, it can be seen that under the same friction conditions, the friction coefficients of ST are higher than those of ST/CE (Ni) and untreated base material (1.353) which shows that the wear resistance is obviously improved after pure nickel plating, at same time, the friction coefficient of textured graphene/nickel composite coatings decreases gradually with the increase of graphene content. The friction coefficient reaches the minimum and stabilize at 0.735 when graphene content is 1.5 mg with decrease rate 29.74% and 45.67% by comparing with ST/CE(Ni) and untreated 6065Al.

Round 4

Reviewer 4 Report

The work discussed the subject of the Preparation and Properties of Graphene/Nickel Composite Coating Based on Textured Surface of Aluminum Alloy. Presented in the manuscript research shows a low level of significance at the present form and from the above reason and taking care of the high level of research target to Materials journal, I recommend rejecting or major revision of this article.

The previous remarks were not taken seriously into consideration by the authors. No meaningful changes were introduced into the work to provide proper material characterization. The authors discussed the effects of proposed treatment without, however, a proper judgement of the material characteristics by itself. Above approach in the reviewer, opinion stays unacceptable especially when the manuscript stays considered for publication in Materials journal.

Author Response

Comments and Suggestions for Authors:

The work discussed the subject of the Preparation and Properties of Graphene/Nickel Composite Coating Based on Textures Surface of Aluminum Alloy. Presented in the manuscript research shows a low level of significance at the present form and from the above reason and taking care of the high level of research target to Materials journal, I recommend rejecting or major revision of this article.

The previous remarks were not taken seriously into consideration by the authors. No meaningful changes were introduced into the work to provide proper material characterization. The authors discussed the effects of proposed treatment without, however, a proper judgment of the material characteristics by itself. Above approach in the reviewer, opinion stays unacceptable especially when the manuscript stays considered for publication in Materials journal.

Response: Firstly, we do take remarks seriously into our consideration and try our best to fully address the comments. In the last two revised version, we added hardness variation of substrate after laser processing and wear resistance curve of substrate according to the comments. Furthermore, laser processing is an intermediate process in this paper, which is not the key point of our research. This part will be discussed in future articles.

In this paper, the substrate characteristics were judged by using chemical composition and hardness value. The final state of material characteristics were judged by using SEM image to illustrating the microstructure morphology and electroplating effect, EDS data for defining chemical composition, friction coefficient and wear trace image for wear resistance and Tafel line and electrochemical impedance spectroscopy for corrosion resistance.

I hope this explanation will be considered to illustrate the relevant issues.